# Assessing the Influence of Buried Archaeology on Equine Locomotion Comparison with Ground Penetrating Radar Results

**DOI:** 10.3390/s20102938

**Published:** 2020-05-22

**Authors:** Neil Linford, Russell MacKechnie-Guire, May Cassar

**Affiliations:** 1Historic England, Portsmouth PO4 9LD, UK; 2Centaur Biomechanics, Moreton Morrell CV35 9BB, UK; info@centaurbiomechanics.co.uk; 3Institute for Sustainable Heritage, Bartlett School of Environment, Energy and Sustainable Resources, University College London, London WC1H 0NN, UK; m.cassar@ucl.ac.uk

**Keywords:** ground penetrating radar, equine kinematics, archaeological remains, high speed video analysis, inertial measurement units, near-surface geophysics

## Abstract

The aim of this trial project was to identify whether buried archaeological remains may have an influence on equine locomotion, through comparison with a non-invasive Ground Penetrating Radar (GPR) survey. This study was conducted at the world-renowned Burghley Horse Trials site, near Stamford, City of Peterborough, U.K. that has a diverse range of heritage assets throughout the wider park land centred on the Grade 1 listed Elizabethan Burghley House. The initial aim of the research was to first use geophysical survey to identify and characterise archaeological remains, and then to determine a suitable location to conduct an equine locomotion study. This trial was conducted with five event type horses with their gaits recorded through the use of three axis, wireless, Inertial Measurement Units, and high speed video capture. It was hoped that this study might indicate an association between the presence of well preserved archaeological remains and changes in the gait of the horses, similar to those shown by studies of dressage horses over different riding surfaces. The results from the equine locomotion study did demonstrate a correlation between the presence of surviving archaeological remains and the alteration in the horses’ gait and, although this is only a preliminary study, the results may well be of interest during the design and construction of equine event facilities. Geophysical survey could, for example, be considered during the design of new or alteration to existing equine courses to allow some mitigation in the location of the course with respect to any archaeological remains, or through the appropriate use of a protective artificial surface.

## 1. Introduction

The aim of this research was to examine the potential influence buried archaeological remains may have on the ridden horse and whether this has a measurable impact on performance or animal welfare when historic assets are present. There is a strong association between the location of equine event courses and historic assets, both at an elite level such as Badminton and Burghley, but also at a more local level. The Institute of Sustainable Heritage, University College London, instigated a research project “*Looking at Old Ground in a New Way: a cross-disciplinary public-private partnership for innovation*” in collaboration with the Burghley Estate and the world renowned Burghley Horse Trials to examine a number of issues regarding the location of the various elements of the event and the support facilities within the park with respect to any surviving archaeological remains. There was specific interest in the location of the competition arena, at the centre of the Horse Trials from the results of a desk top study which suggested the presence of the remains of a medieval monastery on the site.

The characteristics of the riding surface can increase the likelihood of lameness in the dressage horse, particularly over patchy or uneven surfaces, when the firmness of the surface may change compared to uniform conditions [1,2,3]. The sub-surface also has an impact on the way a horse propels its mass forward. In supporting its mass, the surface properties are crucial to optimise the locomotor system, and to give energy back to the horse to aid its locomotion. Some (artificial) surfaces have been designed to be firm and, although there may be some mechanical benefits, there is concern that there is an increase in ground reaction forces and an increased breaking force which over time could lead to injury [4]. In terms of a designed competition course the presence of buried structures, including archaeological remains, could influence the performance of a horse and rider, leading to some calls to introduce more homogeneous artificial surfaces.

The use of artificial surfaces is well established in human sports. Studies have informed the benefits of athletes using a surface in order to reduce the risk of injury and improve training and performance. The physiological adaptions made during training are thought to help protect the athlete against injury. Unlike other sports, where there is a more uniform surface across training/sporting facilities, in equestrian dressage and show jumping it is likely that the surface the horse is trained on would be different from the surface on which they compete. In respect of eventing, it is likely that a horse will be trained on one surface and perform each element on a different surface. This raises significant concerns, given the lack of time available for the horse to physiologically adapt between surfaces. Surface properties can have an effect on the stresses and strains experienced by the horse during locomotion. There is a high prevalence of lameness in the United Kingdom, injuries vary between disciplines (dressage, show jumping and eventing). Given the rise in injury, research has attempted to look at the potential causes which could be derived from training and management [1,5]. This study examines the effect that turf surfaces with underlying archaeology may have on the locomotion of the horse, related to the depth and extent of the buried remains. Hopefully, this novel study will augment other similar research by introducing geophysical survey to investigate the immediate subsurface and take account of presence of archaeological remains where the competition surfaces have not been specially constructed or prepared. This may well suggest a combined approach in both future research and the design and construction of equine competition courses.

Figure 1 shows the location of the Event Arena, initially selected as a site for the study, due to the possible presence of remains form the Priory of St Michael, Stamford, a Benedictine nunnery founded in the mid-12th century and dissolved in 1536. These remains were believed to lie under the current event arena to north of the estate, although this was later found to be incorrect due to an erroneous location record and the absence of any significant geophysical anomalies. The actual remains of the priory are found in Stamford approximately 1.5 km to the west of the estate. The study was therefore conducted on a second site situated over the presumed course of the Ermine Street Roman road to the south of the site in vicinity of the Cottesmore Leap jump on cross country course. Ermine Street was a major Roman road that ran from London and Lincoln built between 45 and 75AD.

## 2. Materials and Methods

### 2.1. Ground Penetrating Radar Survey

The presence of archaeological remains was initially assessed from known sources recorded in the local Historic Environment Record (HER) and the National Heritage List for England (NHLE). A geophysical survey, using a Ground Penetrating Radar (GPR) array was used in advance to accurately locate and characterise significant archaeological remains to determine the best site to conduct the motion analysis study. The vehicle towed GPR survey (Figure 2) was conducted with a MkIV GeoScope Continuous Wave Step-Frequency (CWSF) Ground Penetrating Radar (GPR) system (3d-Radar, Trondheim, Norway) collecting data with a multi-element DXG1820 vehicle towed, ground coupled antenna array [6,7,8]. Data were acquired at a 0.075 m × 0.075 m sample interval across a continuous wave stepped frequency range from 40 MHz to 2.99 GHz in 4 MHz increments using a dwell time of 3 ms. Data processing followed [9].

### 2.2. Equine Locomotion Survey

#### 2.2.1. Kinematics—Inertial Measurement Units

A convenience sample of five event type horses (8 years old) all ridden by their associated rider (five female riders) was used for the study. Horses were in regular work preceding the study and were deemed fit to perform their usual duties. Participation of horses was voluntary and the participants could withdraw from the study at any time. Horses were instrumented with five MTw inertial measurement units (IMU) (Xsens). These were attached over the poll, wither, sacrum, left and right tuber coxae, using custom built pouches and double sided tape (Figure 3). Sensor data were collected at 60 Hz per individual sensor channel and transmitted via proprietary wireless data transmission protocol (Xsens), to a receiver station (Awinda, Xsens, Enschende, Netherlands) connected to a laptop computer running MTManager (Xsens) software (Figure 3).

Two experimental tracks were laid out using a Trimble R8 GNNS (Trimble, Sunnyvale, CA, USA). Track 1 was set up over an area of pre-determined archaeology confirmed by the results of the GPR survey. The test track was 1.5 m wide and 28 m long and marked by spherical cones to identify the track to the riders. A second track of the same dimensions was set up over ground which did not have any significant archaeology. Neither track was located on the current cross- country course. Horses were ridden by their associated rider and three repeats on the left and right rein were collected in trot and canter on both surfaces. If the horse lost straightness, tripped or made an obvious alteration in gait pattern (e.g., shying) the trial was repeated.

IMU data were processed following published protocol. In brief: tri-axial sensor acceleration data were rotated into a gravity (z: vertical) and horse-based (x: craniocaudal and y: mediolateral) reference frame and double integrated to displacement. Displacement data were segmented into individual strides based on vertical velocity of the sacrum sensor and average values for the following kinematic variables were calculated over all strides for each exercise condition:Range of motion: maximum–minimum value over a stride cycle for x, y and z for trot and canter.Minimum difference (MinD_iff_): difference between the two minima in vertical (z) displacement observed during the two diagonal stance phases in trot.Maximum difference (MaxD_iff_): difference between the two maxima in vertical.Displacement observed after the two diagonal stance phases in trot.

IMU derived kinematic variables were summarized between reins. Range of motion variables were subtracted from each other (left rein value − right rein value) and movement symmetry values (MinD_iff_ and MaxD_iff_) were added up (left rein value + right rein value). This procedure ensured that for horses performing symmetrically between reins, values near zero are expected, since head and pelvic movement symmetry values show directional circle dependent tendencies (positive for one rein, negative for the other) and ranges of motion would be expected to be near identical for the left and right rein.

#### 2.2.2. Kinematics—2-Dimensional Motion Capture

Kinematic data were recorded with a high-speed video camera system, using nine markers (30 mm) placed on each horse using double sided tape. Marker locations were identified by manual palpation of anatomical landmarks identifying joint centres and segment ends. Markers were located: lateral condyle of humerus; lateral metacarpal condyles; distal aspect of the metacarpus over the lateral collateral ligament of the metacarpophalangeal joint; lateral condyle of the femur; talus; and distal aspect of the metatarsus over the lateral collateral ligament of the metatarsophalangeal joint. Data were collected from both the left and right rein with three repeats per direction.

One high speed camera (Quintic Biomechanics, Birmingham, UK) was positioned at a ten metre distance from the experiment track, capturing left and right sides of the horse at 400 Hz (spatial resolution 1300 × 400, 400 fps at 10 m distance), with a field of view capturing two complete strides in trot and canter. A halogen light was used to illuminate the markers. High speed video data were recorded and downloaded to a laptop and processed using two dimensional motion capture (Quintic, Birmingham, UK). This experimental technique has been described previously. Automatic marker tracking was used to investigate maximum carpal flexion (palmar angle between lateral condyle of humerus, lateral metacarpal condyles and distal aspect of the metacarpus over the lateral collateral ligament of the metacarpophalangeal joint, maximum tarsal flexion (angle between lateral condyle of the femur, talus, and distal aspect of the metatarsus over the lateral collateral ligament of the metatarsophalangeal joint) during the swing phase. All raw data were smoothed using a Butterworth low-pass filter with a cut off frequency 10 Hz [10].

## 3. Results

### 3.1. Ground Penetrating Radar Survey

The GPR survey over the site of the main competition arena (2.2 ha), close to Burghley House itself, revealed anomalies most likely associated with the infrastructure for the horse trials spectator stands, including a large number of service runs [10,11]. Historic mapping evidence for wider tree planting across the site and small scale quarrying is also, potentially, found in the data (Figure 4).

From approximately 20 ns (1.07 m) onwards a series of low amplitude ditch type anomalies appear to be distinct from the near-surface services and suggest either a geomorphological or, perhaps, more significant archaeological origin. It is difficult to suggest a more complete interpretation due to the concentration of modern services and the possibility these may be related to ridge and furrow agricultural patterns recorded from aerial photography in the immediate vicinity. The results do not indicate the presence of any significant structural remains such as a monastery on the site. More significant results were found over the course of the Ermine Street Roman road, in the vicinity of the Cottesmore Leap on the cross country eventing course. The survey here (1.6 ha) revealed a well-preserved section of Roman road, although the survival seems compromised by ploughing to the east of Queen Anne’s Avenue. The GPR data suggested the location of two test tracks one situated over the remains of the Roman road and one immediately north of the road with little or no apparent archaeology present (Figure 5).

The response to the Roman road is initially defined by a linear anomaly possibly the southern ditch or metalled pedestrian way, which is described across the full extent of the survey area, before the main agger carriageway becomes apparent from between 5.0 and 20.0 ns (0.27 to 1.07 m). The metalled agger is approximately 5 m wide and appears to be much better defined over a short 30 m course immediately to the west of Queen Anne’s Avenue, corroborating a similar response seen in recent parch marks. Deeper reflections between approximately 20 and 40 ns show the course of the road gradually fading to a series of narrow central linear anomalies, with a slightly broader linear response defining the northern extent (Figure 6).

Figure 7 shows a series of low amplitude ditch type anomalies (D) are found from approximately 10 ns (0.54 m) onwards to the north of the Roman road and may well represent a different phase of prehistoric enclosure activity at the site. Broader ditch type anomalies (G) are also found to the south of the road in the deeper data. It is difficult to fully interpret as these anomalies are only partially described within the survey area, although the depth and slightly angular morphology may, possibly, be suggestive of geomorphological features. Certainly, the wide low amplitude anomaly (V) crossing beneath the road between seems likely to represent a dry valley or former water course following a topographic depression sharing the same orientation.

The response to the Roman road immediately to the east of Queen Anne’s Avenue is more subdued, appearing as narrow linear anomalies (BB’) with no indication of any surviving metalled surfacing (Figure 6B). Additional trial survey conducted 350 m further to the east, in a level field where a straight linear projection of the Roman road suggests it should pass, produced no convincing anomalies beyond an ill-defined linear response. It seems likely that the survival of the Roman road to the east of Queen Anne’s Avenue has, perhaps, been compromised by recent mechanical ploughing in these former arable fields or, potentially, the route veered to the south under the modern road to maintain a more level course around the uneven topography found here A more complete discussion of the geophysical survey data can be found in [11].

### 3.2. Equine Locomotion Analysis

As a horse’s hoof makes contact with the ground surface it undergoes a transfer of energy through a series of phases: initial hoof landing and braking (touchdown), support phase (full contact and load), and finally rollover (take off) when, depending on the nature of the surface, the horse will exert energy to create forward motion [12,13,14]. The ideal surface will provide some deformation, to cushion and absorb energy during the landing phase, but provide sufficient firmness to allow the horse to confidently exert energy to push against the ground during take-off. The horse demonstrates an extraordinary sensitivity to the ground surface conditions which may significantly alter the transfer of energy mid stride, which may alter performance under competition conditions as the surface of event courses and competition arenas wear.

The initial gait analysis experiment used high speed video motion capture to measure the relative movement of each horse’s legs from a series of reflective markers placed on anatomical muscular skeletal landmarks. Automatic marker tracking was used to investigate maximum carpal flexion (palmar angle between lateral condyle of humerus, lateral metacarpal condyles and distal aspect of the metacarpus over the lateral collateral ligament of the metacarpophalangeal joint), maximum tarsal flexion (angle between lateral condyle of the femur, talus, and distal aspect of the metatarsus over the lateral collateral ligament of the metatarsophalangeal joint) during the swing phase. All raw data were smoothed using a Butterworth low-pass filter with a cut off frequency 10 Hz [5]. Table 1 shows the data collected from all five horses in the trial, separated between the experimental track with and without archaeology present, and approaches each track from both the left and the right. No significant differences between the two tracks were apparent from this data set determined from a paired *T*-test with a significance level set at *p* ≤ 0.05.

Table 2 shows the results from the IMU data following statistical analysis performed in SPSS (vers. 22, IBM, Armonk, NY, USA). Kinematic outcome parameters were assessed for normality using a Shapiro Wilks Test and found to be normally distributed. A mixed model was used to determine the influence of speed on outcome parameters. Differences in range of motion in a craniocaudal (x), mediolateral (y) and vertical (z) direction for the wither, sacrum, left and right tuber coxae between the two surfaces were assessed using a paired *T*-test with a significance level set at *p* ≤ 0.05. Since many kinematic parameters are influenced by speed, differences in speed between different conditions was tested. No significant difference was found in any of the outcome parameters when speed was included in the mixed model.

Results from the IMU data, show that when horses were travelling over ground which had underlying archaeology this was associated with an alteration in gait when compared to ground where the archaeology was absent. A variation in gait was also found when the horses were travelling from the right of Track 1, commencing over the best preserved portion of the Roman road agger, possibly suggesting a greater degree of firmness and more uniform surface for the horses to propel their mass, as opposed to an irregular surface which would dampen the ground reaction forces being generated. The findings being presented here warrant further investigation as in some parameters when horses were travelling on track one, some of the symmetry parameters show increased symmetry (poll Min_Diff_ values closer to zero representing symmetry) however, when going in the opposite direction (right rein), these asymmetry parameters showed increased asymmetry. This could be explained by the horse’s biological adaption to the surface and or equine laterality with directional preferences.

### 3.3. Post Equine Locomotion Analysis GPR Survey

No invasive means of determining the firmness of the ground was possible in this trial study, but a repeat GPR survey was conducted immediately after the equine motion analysis trial which provided useful information regarding very near-surface ground deformation due to the horses’ locomotion over the experimental tracks.

The ground surface showed obvious deformation along the centre of each track from the repeat, and equal, movements of the five horses over both Track 1 and Track 2. As the precise geophysical signature of this surface deformation was unknown, a simplified model [15,16,17] based on a series of air-filled depressions representing hoof marks was, with varying widths (0.1 to 0.2 m) and depths from 0.05 to 0.2 m from the ground surface, was constructed (Figure 8). Analysis of the model data suggests the most prominent reflections occur from the air/soil interface at the base of each individual depression. More complex reflections are found when the individual depressions are modelled as polygons with sloping faces or where close neighbouring objects physically coincide. As would be expected a correct estimate to the base of each depressions is given by assuming a velocity of the radar wave front in air (0.29 m/ns) rather than the background soil host medium (0.1 m/ns).

From the results of the numerical model, the influence of any anomalies due to the surface deformation caused by the horses is unlikely to extend beyond ~2 ns and should only be visible in the very near-surface time slice images. Figure 9 shows field GPR data collected immediately after the equine locomotion analysis experiments and shows the anomalous response to the surface wear along Track 1 appears between 1.0 and 2.0 m, which would suggest a depth to the top of any air-filled depressions of approximately 0.15 m.

As might be expected the GPR does not, necessarily, have sufficient resolution to determine the precise extent of individual depressions caused by surface wear due to the horses or determine the depth from the surface. This might be achieved more readily in future work through use of a Terrestrial Laser Scanner. However, it is of interest to note that the GPR data has detected anomalies associated with surface wear which correlate with greater concentration of near-surface archaeological remains found to the east of Track 1. This, together with the absence of similar anomalies over Track 2, perhaps suggests an interaction between the horse over the near-surface archaeological remains resulting in a greater degree of surface wear.

## 4. Discussion

Results from the GPR survey successfully located and mapped the course of the Ermine Street Roman road in the vicinity of the Cottesmore Leap cross country fence. The underlying archaeology is relatively shallow, producing detectable anomalies from approximately 0.13 m from the ground surface extending to a depth of 1.74 m. The apparent survival and depth extent of the Roman road varies considerably within the survey area with the most prominent remains found immediately to the west of Queen Anne’s Avenue which was therefore chosen for the motion analysis test site. In addition, there is a variation in the depth and concentration of the archaeological remains from east (right) to west (left) along the course of Track 1 as this is located partially on the apparently well preserved main agger surface of the Roman road. Results from the wider survey area also revealed additional significant anomalies potentially related to a system of ditches and what appears to be the response from geomorphological features.

Whilst the results from equine kinematics pilot study are limited by the sample size of five horses, a correlation between the changes in equine gait and the presence of buried archaeological remains has been demonstrated and warrant further investigation. Some direct parallels are suggested between the relative survival of the archaeological remains and observations made on the condition of artificial equine sports surfaces. In essence the horse is able to stabilise and propel their mass more efficiently when the ground has significant archaeology present. Also partially reflected in the greater degree of track wear evident where the underlying Roman road created a firmer riding surface. The location and survival of archaeological remains may, if unrecognized, alter the gait of a horse on approach to a fence or obstacle providing an element of variability inherent to an individual course. This does not, necessarily, equate to an advantage to the horse and rider as the buried remains may result in a rapid variation in the firmness of the ground and could, under certain circumstances, increase the risk of a fall should the horse sense changes in the stability of the ground that are not apparent to the rider. Further similar studies would help validate these initial results and might also include comparisons with artificial event arena surfaces, where the changes of equine gait found here should be less pronounced. The influence of the depth to the archaeological remains might also be investigated through similar studies conducted over a wider range of site conditions.

## Figures and Tables

**Figure 1 sensors-20-02938-f001:**
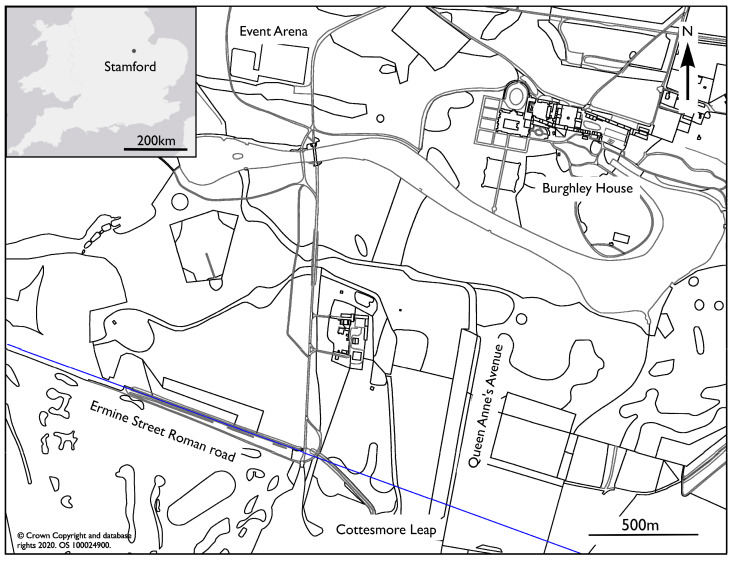
Location of the study site near Stamford, Lincolnshire, UK (inset) together with a plan of the Burghley Estate. The equine locomotion study was conducted over a test site in the vicinity of the Cottesmore Leap to the south of the estate where the presumed course of the Ermine Street Roman road is marked by a blue line.

**Figure 2 sensors-20-02938-f002:**
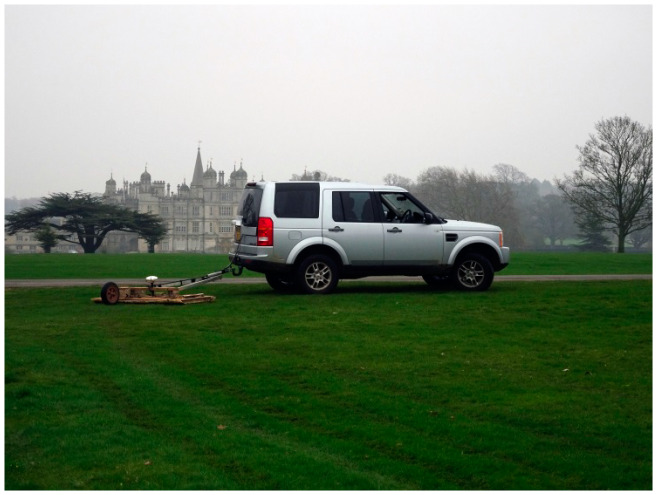
Vehicle towed ground penetrating radar survey in progress over the site of the competition arena with Burghley House in the background. The 20 channel GPR array is towed behind the vehicle with measurements triggered every 0.075 m by the odometer wheel and positional control provided by an RTK GNSS receiver mounted above the centre of the GPR array.

**Figure 3 sensors-20-02938-f003:**
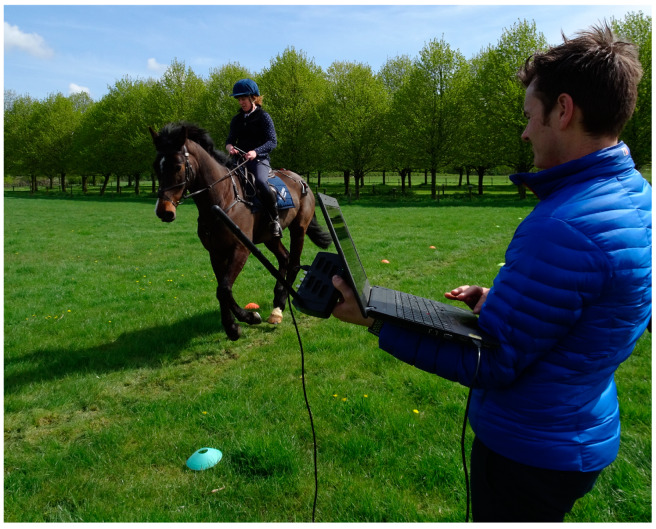
Equine motion analysis in progress. The horse has been instrumented with 5 triple axis inertial measurement units to monitor any changes in gait when passing over the location of the Ermine Street Roman road. The position of the test track determined from the previous geophysical survey is marked on the ground by the coloured cones.

**Figure 4 sensors-20-02938-f004:**
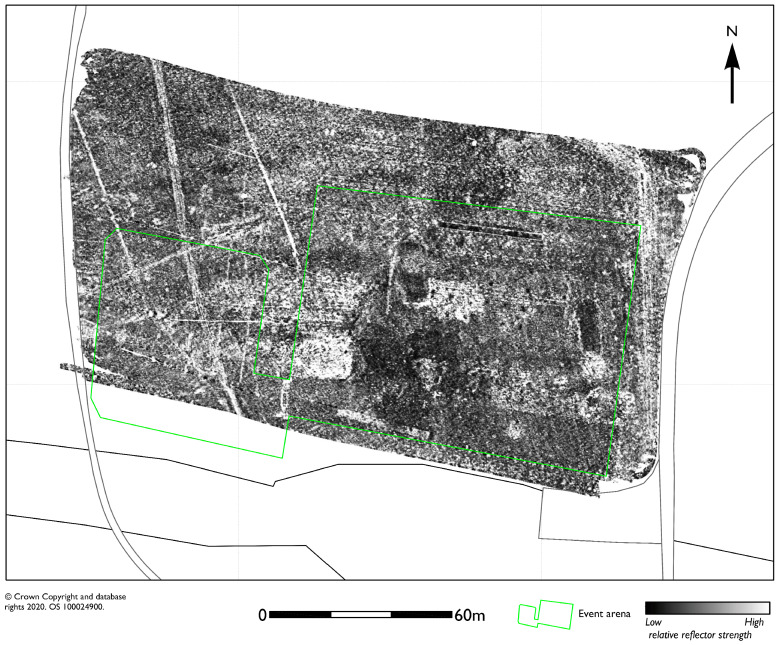
GPR amplitude timeslice from between 5.0 and 7.5 ns (0.27 to 0.4 m) showing a near-surface response dominated by utilities and other infrastructure associated with the competition arena, set out within the area marked in green.

**Figure 5 sensors-20-02938-f005:**
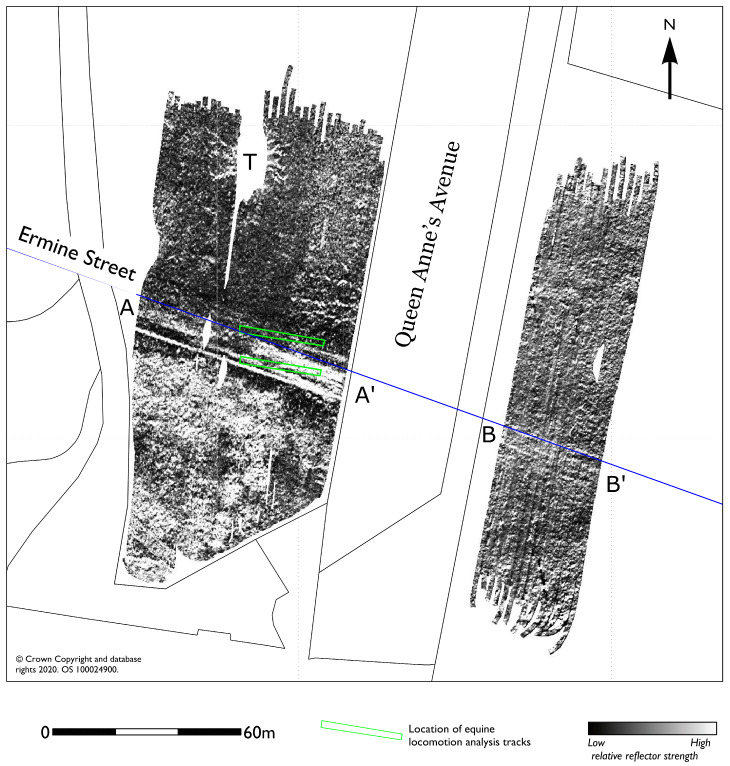
GPR amplitude timeslice from between 5.0 and 7.5 ns (0.27 to 0.4 m) showing the course of the Ermine Street Roman road (blue line) that survives better in the park land (AA’) to the west of Queen Anne’s Avenue, than in the field to the east (BB’) which has been subject to a more intense arable regime in recent years before being put down to grass. The two experimental tracks for the equine locomotion analysis are shown by the green boxes. Some gaps in the GPR coverage were necessary due to the presence of mature trees (T) where the root systems have also been imaged by the GPR survey.

**Figure 6 sensors-20-02938-f006:**
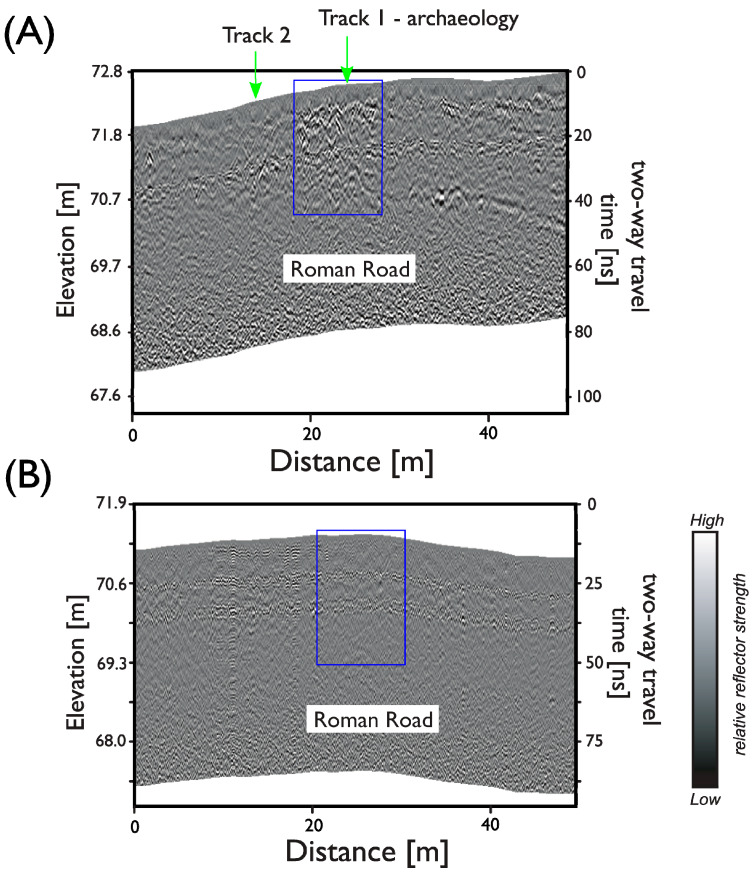
GPR profiles over (**A**) the well-preserved section of Roman road (shown by a blue box) to the west of Queen Anne’s Avenue and (**B**) in the former arable field to the east. The location of the two equine locomotion analysis test tracks is also shown by the green arrows in (**A**) where the depth from the ground surface to the anomaly due to the Roman road is approximately 0.25 m.

**Figure 7 sensors-20-02938-f007:**
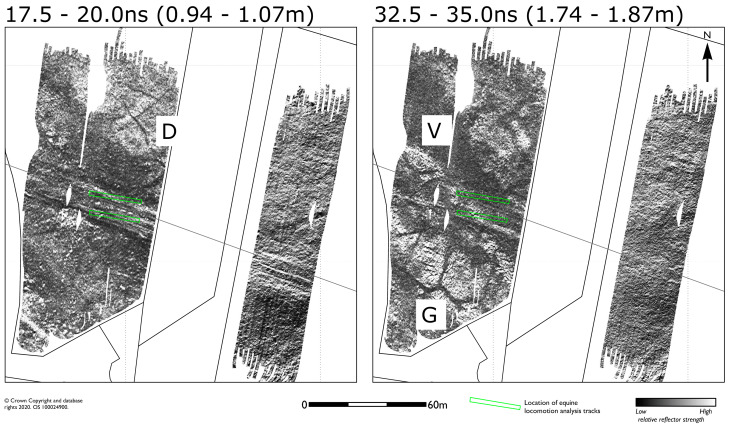
GPR amplitude timeslices from between 17.5 and 20.0 ns (0.94 to 1.07 m), and between 32.5 and 35.0 ns (1.74 to 1.87 m) illustrating the variation in survival and approximate depth of the Roman road. A system of enclosure ditches (D) are found to the north of the Roman road, together with possible geomorphological features (G) and a deeper dry valley (V).

**Figure 8 sensors-20-02938-f008:**
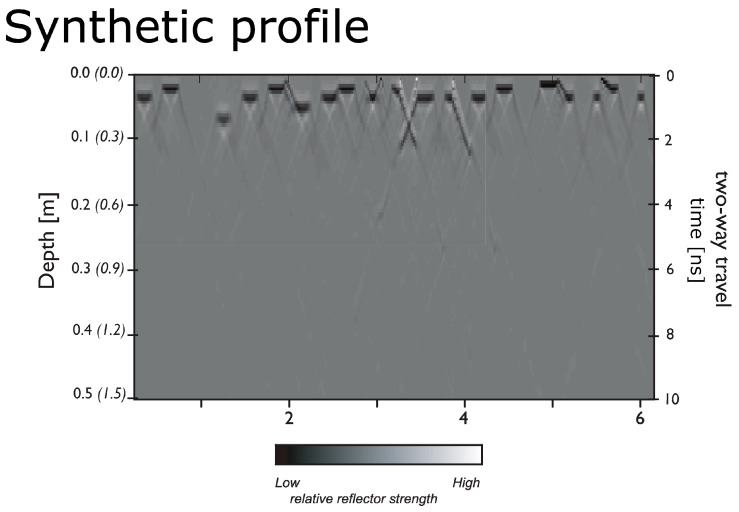
Physical model showing the synthetic GPR profile expected from a series of air-filled hoof depressions in the ground surface.

**Figure 9 sensors-20-02938-f009:**
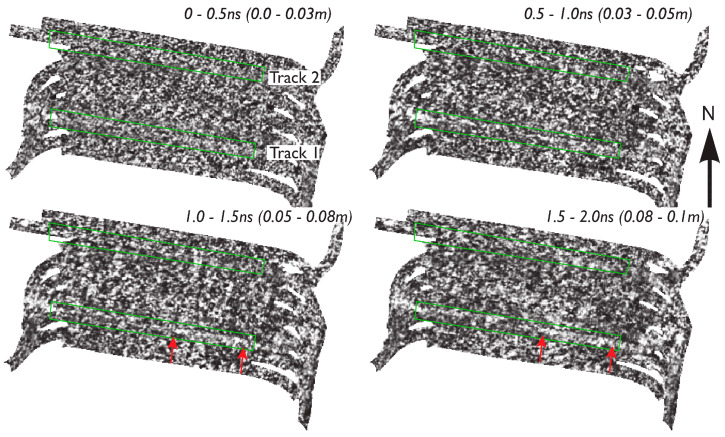
Field GPR data collected immediately after the equine locomotion analysis. The red arrows indicate enhanced wear from the right of Track 1 where the archaeology provides more firm ground conditions, encouraging the horses to transfer more energy during forward propulsion.

**Table 1 sensors-20-02938-t001:** Kinematic data output from the high speed video capture, no significant difference found between the two surfaces.

	Track 1 Archaeology	Track 2	*p* ≤ 0.05
Max Carpal flexion left	91.47°	94.57°	0.16
Max Carpal flexion right	93.87°	92.61°	0.53
Max Tarsal flexion left	110.10°	114.44°	0.55
Max Tarsal flexion right	115.28°	115.23°	0.98

**Table 2 sensors-20-02938-t002:** Kinematic data from the IMU sensors collected during trot and canter for both left and right rein on Track 1 and Track 2, (ROMY = range of motion in mediolateral direction, ROMX = range of motion craniocaudal direction ROMZ = range of motion in vertical direction, MinD_iff_ = difference between the two minima in vertical displacement, MaxDiff = difference between the two maxima in vertical displacement).

Direction	Sensor location	*p* ≤ 0.05	Track 1 Archaeology (mm)	Track 2 (mm)
Trot				
left rein	Poll ROM X	*0.05*	36.6	31.75
left rein	Poll Mindiff	*0.05*	2.05	−5.24
right rein	Sacrum ROM X	*0.03*	18.2	19.8
right rein	RTC ROM Y	0.06	38.6	31.2
right rein	Poll Mindiff	*0.04*	5.91	1.73
right rein	Wither Max_diff_	0.07	9.67	5.51
right rein	LTC Max_diff_	*0.02*	−16.08	−19.24
Canter				
right rein	Sacrum ROM Z	0.06	175.0	185.2
right rein	Wither ROM Y	0.07	57.4	48.6

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
