# Peer review of "Assessing the Influence of Buried Archaeology on Equine Locomotion Comparison with Ground Penetrating Radar Results"

_sensors, 2020, doi:10.3390/s20102938_

Round 1

Reviewer 1 Report

Very interesting article and multidisciplinary. 

Explain: 31-33 the results may well be of interest during the design and construction of equine event facilities. In Discussion 

Author Response

Point 1, Very interesting article and multidisciplinary. 

Response 1, Thank you for your attention to our article and the useful comments you have made.

Point 2, Explain: 31-33 the results may well be of interest during the design and construction of equine event facilities. In Discussion 

Response 2, Additional discussion has been added to this section, as you suggest, to explain how the use of geophysical survey might help in the design and construction of equine facilities. This may largely be envisaged as suggesting minor alterations in the location of the course to take account of the presence or absence of significant remains.

Reviewer 2 Report

Paper focused on a preliminary project to identify whether buried archaeological remains may have an influence on equine locomotion using GPR data acquisition and analysis. ……. A strange research project, but it constitutes an interesting develop related to a correlation between buried archaeological remain and the horse performance. Results are very interesting if well explaned.   

Improvements are request

  • Abstract needs to be more concise, focus on location, need for investigation methods used expected outcomes.
  • Introduction: authors should be clear how your work could be advances the state of the art or the knowledge in this field. I think that the Authors could be focused this paragraph to specify the peculiar aspects of novelty of this paper with respect to other studies performed from other authors.
  • I believe that would be much more interesting if Authors provide two or more 2D radar sections to understand the compactness of the shallow subsurface in the presence of archaeological structures and in the absence of archaeological structures. Can the depth of the burials affect the movement of the horses?

Author Response

Paper focused on a preliminary project to identify whether buried archaeological remains may have an influence on equine locomotion using GPR data acquisition and analysis. ……. A strange research project, but it constitutes an interesting develop related to a correlation between buried archaeological remain and the horse performance. Results are very interesting if well explaned.   

Many thanks for your attention to the manuscript and very useful comments below. This was certainly a very interesting multi-disciplinary project with a lot of really useful outcomes.

Improvements are request

  • Abstract needs to be more concise, focus on location, need for investigation methods used expected outcomes.

Response 1, the abstract has been redrafted to meet your useful suggestions.

  • Introduction: authors should be clear how your work could be advances the state of the art or the knowledge in this field. I think that the Authors could be focused this paragraph to specify the peculiar aspects of novelty of this paper with respect to other studies performed from other authors.

Response 2, this is more difficult comment to address. This is a relatively unique area of research, at present, and was undertaken as a pilot study to test the feasibility of combining non-invasive investigation using geophysics with equine locomotion analysis. It is, perhaps, premature to make wider comparison at this stage although we do emphasise the merits of non-invasive means of investigation.

  • I believe that would be much more interesting if Authors provide two or more 2D radar sections to understand the compactness of the shallow subsurface in the presence of archaeological structures and in the absence of archaeological structures. Can the depth of the burials affect the movement of the horses?

Response 3, This is a very useful suggestion and a new figure (Figure 6) has been included showing two profile sections as suggested. For clarity, these focus on the immediate vicinity of the Roman road comparing sections where it is well preserved against a section where survival appears poor. The question of depth is certainly of interest but beyond the scope of the current study, perhaps a topic to be investigated further in future work.

Reviewer 3 Report

See Comments to Editor.

Author Response

This manuscript focuses on the trial of identifying whether buried archaeological remains influence equine locomotion using Ground Penetrating Radar (GPR) survey near Stamford, England. I am quite familiar with GPR in my own research on archaeological sites but not in combination with sport nor involving horses. The topic is quite fascinating and I found the manuscript to be well researched, well organized, and well written. A clear aim in the Introduction is also well addressed in the Conclusion. I think this manuscript should be published but with after a few minor changes/corrections. I do not think my comments are overly difficult to fix, and mostly pertain to the methodology. As an archaeologist who uses GPR, I am not interested in its utility and how these results can be replicated by other researchers, whether involved in equine locomotion or canine herding trials.

As I mentioned, the manuscript is well written and I only found a couple instances of any grammatical issues (see below). I do have some questions about the research and understand some may be addressed given the word limit. Some questions may relate to me thinking out loud about how and why decisions were made, and also understand if these are not commented on during revisions.

Many thanks for your attention to the manuscript and very useful comments you make below, hopefully we have been able to address these in the revised manuscript and responses included here.

Point 1 In the manuscript, I noticed that there is little to no information describing the Ermine St. Roman Road or the monastery. Both of these need to be addressed for any reader not familiar with either of these features/sites. For example, when were they built (and abandoned) and where on the Burghley House grounds are they located, especially in relation to the House itself. A map is needed first to locate the Burghley House within England, and second where are these two features/sites on the grounds. This map is important before any discussion of the GPR is given. In other words, I am lacking spatial context here.

Response 1 the introduction now has a brief section describing both the monastic remains and the Roman road – this also explains that the priory is not actually found on the estate at all and was mis-located on one of the historic environment records. The suggestion of a location plan is really useful and is now included as Figure 1 – hopefully this provides all of the spatial information you suggest would be useful?

Point 2 Do the current owners of Burghley House know of these buried features? I presume they must have if they are listed on the National Heritage List. I am asking because it may be relevant to note that the owners knew this and therefore asked for this trial project to be undertaken. I am curious here how the project originated (maybe I missed this in the manuscript).

Response 2 We provide some background discussion to the project in the introduction that I think covers the points you raise here? This was a fully collaborative project supported by all the stake holders involved.

Line 93: space between competition and arena.

Point 3 The number of horses involved in the study is listed first in the Abstract but not until much later in the paper. I think it needs to be mentioned in the Materials and Methods section as well. Also, how many times did the horses go over Tracks 1 and 2? In other words, how many times did the horses make the jumps on both tracks, and does the number matter after passing a certain threshold?

Response 3 we have added more information to the method section, as you suggest, including details of the number of repeats over each track. Note that the tracks did not include a fence to jump and results from each horse was aggregated to a mean for each horse. Point 4 Line 155: This is the first time (other than the Abstract) that Ermine St Roman Road is mentioned. Again, this information needs to be presented much earlier, along with that of the monastery. These should not be afterthoughts given they are the only archaeological features in the study area.

Line 160: I have no real sense of how far away these two archaeological features are from the House itself. The map of the grounds and where the test tracks are in relation to modern structures will help immensely. And, could you state how many anomalies there were, other than using the word ‘plethora’.

Response 4 hopefully this is now covered in the introduction / location map? The results section also provides a brief description of the anomalies in the Event Arena (mainly modern services) and explains the mis-location of the monastic site.

Point 5 I know there was no digging to ground-truth the anomalies but perhaps this could be stated so the reader knows not to expect such information. I do understand that by its very nature a GPR study is non-invasive but want to clear no archaeological testing was performed.

Response 5 that is correct this was a non-invasive study only.

Point 6 Line 170: Having the map of the grounds will bolster the notion in this line that no structural remains of the monastery was found.

Lines 175-176: Again the Ermine St Roman Road is discussed but I have no idea where to locate that based on Figure 3. I know in Figure 4 Queen Anne’s Avenue is shown but it is not in Figure 3 so cannot see how the Roman Road and Queen Anne’s Avenue are related. How does the monastery fit into these figures? And, the same for Cottesmore Leap? It is mentioned for the first time in Line 320. If it is a modern fence this needs to be stated and where it is related to other historic and modern features on the grounds.

Response 6 hopefully all of the spatial queries should be covered on Figure 1 with labels for all of the sites mentioned in the text. The actual recorded site of the monastery is not included in this figure as it lies too far to the west of the Burghley estate.

Point 7 Figure 4: the green lines need to have a thicker weight or a different color to stand out more.

Line 193: Is ‘a more intense arable regime’ referring to modern or historic usage? On Line 212 you say ‘former’ arable fields so I want to make sure this means historic and not still in modern use.

Response 7 green lines have been made more prominent and the course of the Roman road is now marked more clearly too. This is recent mechanical plough degradation and both figure caption and text have been updated to clarify this useful point.

Point 8 Line 199: Is a ‘different phase of enclosure activity” referring to modern or historic use? Orm just not Roman in date?

Response 8 most likely prehistoric now updated with this interpretation in the text.

Line 213: need period after ‘here’.

Point 9 Line 288: How do the authors know the movements recorded were from the 5 horses in the current study and not also from previous events? Was the course closed for a certain duration of time for the soil to rebound before they tested the course? If closed then what was that duration? Just inquiring. I may have missed this earlier on.

Response 9 hopefully now clarified in the text but both test tracks were located in an area that was not on the current cross country course, so had not been used by any other riders.

Line 329: need period after ‘features’.

Point 10 Line 336: So, this means that Track 1 is better for the horse because of the presence of the buried archaeological features. My understanding then is that horses will perform better where these buried features are located – and by that you mean that artificial surfaces are better surfaces for the horse because, in both instances, they provide the kind of stability the horse needs. I think it would be important to list the takeaways here so that it is clear to the reader what the outcomes are from the authors. Clearly, if buried archaeological features are better for the health of the horse during these jumping events and these buried features mirror/mimic artificial surfaces then it is important to state it with emphasis.

Response 10 hopefully now covered more fully in the discussion section, it is the variability in the ground conditions that is of more concern than the presence or absence of the archaeology. An artificial surface may provide more even conditions reducing variability during a competition and avoiding sudden changes of the subsurface that may result in horse changing their expected gait on the approach to a jump.

Point 11 One last question comes to mind and that relates to whether there are existing artificial surfaces being used in equine sporting events around the country and/or world. Has a study like this ever been performed on artificial surfaces elsewhere? If so, list those for comparative purposes and discuss those results. Is this study the first of its kind? If so, then the takeaways become even more relevant to include?

Response 11 This study is fairly unique but also only a pilot so we would hesitate to draw too many conclusions before there has been an opportunity to replicate and the extend the work with similar studies. Artificial surfaces are used in some settings, for example in the temporary event arena constructed for the 2012 London Olympic games, and we include relevant references to similar studies (for dressage horses).

Point 12 Overall, my comments are relatively minor. I do think more descriptive information needs to be included about the archaeological features presented. And, they need to be placed on a map so the reader can see where they are located in relation to modern structures. As a manuscript focused on spatial information I think this information needs to be included for clarity. As I stated at the beginning, this is an interesting paper and it would be useful for the authors to discuss some of the pertinent takeaways in the Conclusion, including how this kind of research can be applied to other equine events across the country/world, to other sporting events of a similar nature (dogs for example), to non-equine sporting events. What about its applicability to steeplechase runners? This goes to one of my initial comments about the utility of GPR in such projects that are not strictly archaeology focused – and studying equine locomotion is not typically what GPR was designed for so I applaud its application elsewhere. The manuscript is publishable with these minor comments in mind.